# Assessment of Renal Dysfunction Improves the Simplified Pulmonary Embolism Severity Index (sPESI) for Risk Stratification in Patients with Acute Pulmonary Embolism

**DOI:** 10.3390/jcm8020160

**Published:** 2019-02-01

**Authors:** Antonin Trimaille, Benjamin Marchandot, Mélanie Girardey, Clotilde Muller, Han S. Lim, Annie Trinh, Patrick Ohlmann, Bruno Moulin, Laurence Jesel, Olivier Morel

**Affiliations:** 1Pôle d’Activité Médico-Chirurgicale Cardio-Vasculaire, Nouvel Hôpital Civil, Centre Hospitalier Universitaire, Université de Strasbourg, 67091 Strasbourg, France; benjamin.marchandot@chru-strasbourg.fr (B.M.); melanie.girardey@chru-strasbourg.fr (M.G.); annie.trinh@chru-strasbourg.fr (A.T.); patrick.ohlmann@chru-strasbourg.fr (P.O.); laurence.jesel@chru-strasbourg.fr (L.J.); olivier.morel@chru-strasbourg.fr (O.M.); 2Pôle NUDE, Nouvel Hôpital Civil, Centre Hospitalier Universitaire, Université de Strasbourg, 67091 Strasbourg, France; clotilde.muller@chru-strasbourg.fr (C.M.); bruno.moulin@chru-strasbourg.fr (B.M.); 3Department of Cardiology, Austin and Northern Health, Melbourne 3084, Australia; hanslim@gmail.com; 4Laboratory of Regenerative Nanomedicine, UMR 1260, INSERM (French National Institute of Health and Medical Research), FMTS (Fédération de Médecine Translationnelle de l’Université de Strasbourg), Faculté de Médecine, Université de Strasbourg, 11 rue Humann, 67085 Strasbourg, France

**Keywords:** chronic kidney disease, cardio-renal syndrome, contrast-induced nephropathy, venous thromboembolism

## Abstract

Background: Whereas the major strength of the simplified pulmonary embolism severity index (sPESI) lies in ruling out an adverse outcome in patients with sPESI of 0, the accuracy of sPESI ≥ 1 in risk assessment remains questionable. In acute pulmonary embolism (APE), the estimated glomerular filtration rate (eGFR) can be viewed as an integrate marker reflecting not only previous chronic kidney disease (CKD) damage but also comorbid conditions and hemodynamic disturbances associated with APE. We sought to determine whether renal dysfunction assessment by eGFR improves the sPESI score risk stratification in patients with APE. Methods: 678 consecutive patients with APE were prospectively enrolled. Renal dysfunction (RD) at diagnosis of APE was defined by eGFR < 60 mL/min/1.73 m^2^ and acute kidney injury (AKI) by elevation of creatinine level >25% during in-hospital stay. Results: RD was observed in 26.9% of the cohort. AKI occurred in 18.8%. A stepwise increase in 30-day mortality, cardiovascular mortality and overall mortality was evident with declining renal function. Multivariate analysis identified RD and CRP (C-reactive protein) level but not sPESI score as independent predictors of 30-day mortality. AKI, 30-day mortality, overall mortality, and cardiovascular mortality were at their highest level in patients with eGFR < 60 mL/min/1.73 m^2^ and sPESI ≥1. Conclusion: in patients with APE, the addition of RD to the sPESI score identifies a specific subset of patients at very high mortality.

## 1. Introduction

Early stratification in patients with acute pulmonary embolism (APE) remains a challenging issue, and there is an ongoing need for the identification of additional markers that can improve the predictive ability of current risk stratification schemes. Among various prediction models, the pulmonary embolism severity index (PESI) allows reliable assessment of 30-day outcome of patients with APE and performed better than the older Geneva prognostic score [1,2]. In an attempt to overcome the complexity of the original PESI [3], a simplified pulmonary embolism severity index (sPESI), was developed and validated, including in the Japanese population. [4] The sPESI focused on 6 equally weighted variables—age >80 years, cancer, chronic heart failure or chronic pulmonary disease, systolic blood pressure < 100 mmHg, arterial oxyhemoglobin saturation < 90% [5]. Previous studies demonstrated that sPESI performed better than the shock index and was at least as accurate as a strategy combining imaging and laboratory parameters for excluding patients at elevated risk [1,5,6]. Although its major strength lies in excluding an adverse outcome in patients with sPESI of 0, the accuracy of sPESI ≥1 in risk assessment remains questionable. Among various biochemical markers that have been proposed as an alternative tool for risk stratification, recent studies have underscored the value of renal dysfunction (RD) at diagnosis or of acute kidney injury (AKI) during hospital stay on the prognosis of patients with APE [7,8,9,10]. In addition to its effects on pulmonary circulation, hemodynamic compromise may induce decreased cardiac output, hypoxemia and elevated central venous pressure that reduces glomerular filtration and favors kidney injury. At the chronic phase, recent data have underlined that balloon pulmonary angioplasty has multiple beneficial effects including kidney function improvement [11]. On the other hand, several registries have underlined a higher prevalence of PE or venous thromboembolism in chronic kidney disease (CKD) patients [9,12,13].

In the present study, we sought to determine whether RD assessed by estimated glomerular filtration rate (eGFR) at diagnosis improves the sPESI score for risk stratification in patients with APE. In addition the impact of acute kidney injury defined as a > 25% increase in creatinine levels during hospital stay on adverse outcome was investigated.

## 2. Methods

This study retrospectively included all eligible patients diagnosed as having APE who were consecutively admitted to the cardiology department of a university hospital (Nouvel Hôpital Civil, University of Strasbourg) from 1 April 2008 to 31 December 2013.

The hospital database was analyzed using the International Classification of Diseases codes (126.0 for acute pulmonary embolism with heart failure, 126.9 for acute pulmonary embolism). Among 797 patients with APE as a primary or secondary diagnosis, 119 patients with APE considered as the secondary diagnosis were excluded. A total of 678 consecutive patients were hospitalized for PE, with the diagnosis confirmed mainly by computed tomographic (CT) pulmonary angiography. Baseline demographics and clinical characteristics were obtained from the medical records. Twelve-lead electrocardiogram (ECG) and routine laboratory data including quantitative Troponin I (TN-I), BNP, C-reactive protein, urea, creatinine, eGFR, pH, PaO_2_, PaCO_2_, D-dimer and leukocyte levels were obtained serially. For each patient, the sPESI at the time of hospital admission was calculated based on the clinical and demographic characteristics of each patient. The flow chart is shown in Figure 1.

### 2.1. Staging of Renal Function

Baseline serum creatinine levels were assessed at diagnosis for all patients. The estimated glomerular filtration rate (eGFR) was calculated using the abbreviated Modification of Diet in Renal Disease (MDRD) formula. Patients were divided into 3 subgroups according to their eGFR levels: Group 1 ≥ 60 mL/min/1.73 m²; Group 2 ≥ 45 and < 60 mL/min/1.73 m²; Group 3 < 45 mL/min/1.73 m²). Patients with an eGFR < 60 mL/min/1.73 m² were assigned to the RD group, whereas those with an eGFR ≥ 60 mL/min/1.73 m² were assigned to the no-RD group. For each patient, the type of renal failure (acute vs. chronic) was established by a careful reviewing of medical and biological records. Acute kidney injury (AKI) during hospital stay was defined by a ≥25% elevation of creatinine levels compared to baseline creatinine at diagnosis. Pre-existing chronic kidney disease (CKD) was determined at the time of enrolment for each patient: (i) according to current KDIGO definitions: kidney damage or glomerular filtration rate (GFR) < 60 mL/min/1.73 m^2^ for 3 months or more, (ii) by evaluating available clinical and laboratory data from the patient or surrogate, (iii) using serum levels of calcium, phosphate, parathormone, alkaline phosphatase activity and albuminuria when available.

A diagnosis of APE was confirmed by a high-probability ventilation-perfusion scan result (according to the criteria of the Prospective Investigation of the Pulmonary Embolism Diagnosis), or APE diagnosed on contrast-enhanced PE-protocol helical computed tomography of the chest. In some patients, APE diagnosis was based on clinical examination and biological data compatible with APE diagnosis (D-dimers elevation, hypoxemia, hypocapnia…) + findings of a lower limb venous compression ultrasonogram positive for a proximal deed vein thrombosis.

### 2.2. Endpoints

The primary endpoint of the study was defined as all-cause mortality at 30 days.

The secondary endpoints were (i) all-cause mortality during follow-up and (ii) cardiovascular death during follow-up.

After hospital discharge, follow-up was obtained using standardized telephone interviews with a cardiologist or another physician. In case of death, the cause was ascertained by thorough review of all available clinical information at the time of death. Cardiac death was defined as any death with demonstrable cardiac cause or any death that was not clearly attributable to a non-cardiac cause.

The study protocol was approved by the Institutional Review Board of the University of Strasbourg. Retrospective consent was obtained from patients when alive at follow-up or from their relatives.

### 2.3. Statistical Analysis

Categorical variables were expressed as count and percentages. Continuous variables were reported as mean ± standard deviation (SD) or as median and interquartile range (25th-75th) according to their distribution. Categorical variables were compared with chi-square test or Fisher’s exact test. Continuous variables were compared with the use of analysis of variance (ANOVA) and Bonferonni correction or non-parametric Kruskal–Wallis as appropriate. To determine predictors of death, Cox-regression analysis was performed.

In the multivariate analysis, only variables with less than 10% missing values were taking into account. Variables with *p* < 0.05 in univariate analysis were entered into a stepwise ascending multivariate analysis. In addition, variables that were taken into account in the sPESI (age, cancer, chronic heart failure or chronic pulmonary disease, systolic blood pressure <100 mmHg, arterial oxyhemoglobin satuation <90%) or clearly related to the sPESI score (BNP, LVEF <50%, PaO_2_) were not entered into the multivariate analysis model.

Associations between RD, sPESI and occurrence of clinical outcomes were assessed by Kaplan Meier analysis and the log rank test. All tests were 2-sided. A *p* value < 0.05 was considered significant. Calculations were performed using SPSS 17.0 for Windows (SPSS Inc., Chicago, IL, USA).

## 3. Results

A total of 678 consecutive patients were enrolled in the study. As prespecified, the cohort was split into 3 subgroups according to the eGFR measured on admission: Group 1 (*n* = 495) eGFR >60 mL/min/1.73 m², group 2 (*n* = 106) eGFR 45 to 60 mL/min/1.73 m², group 3 (*n* = 77) eGFR <45 mL/min/1.73 m². Baseline demographics, clinical and biological characteristics are described in Table 1. In APE patients, RD at diagnosis was observed in 26.9% of the cohort. As expected, RD patients were older and had multiple comorbidities, including ischemic cardiomyopathy, arterial hypertension, dyslipidemia, hypertension and diabetes mellitus. Pulmonary embolism was diagnosed by computed tomographic pulmonary angiography in 516 patients (76.1%), by ventilation/perfusion scanning in 138 patients (20.3%), and by other methods in 24 patients (3.5%). All patients received standard anticoagulant therapy with intravenous unfractionated heparin (UFH) or a subcutaneous body mass-adjusted dose of low-molecular weight heparin. In patients with eGFR <30 mL/min/1.73 m^2^, intravenous UFH under activated partial thromboplastin time (APTT) control was given. All patients were discharged home with an oral vitamin K antagonist treatment by Fluindione.

### 3.1. Acute Kidney Injury

During hospital stay, AKI defined by a >25% elevation of creatinine levels was observed in 128 patients (18.8%). Univariate and multivariate analyses for the prediction of AKI are shown in Table 2.

By multivariate Cox regression analysis, sPESI was the sole independent predictor of AKI. Surprisingly, the use of computed tomography (CT) scanning as the diagnostic method appeared protective. The impact of AKI on 30-day mortality, overall and cardiac mortality at follow-up is shown in Table 3, Table 4 and Table 5.

### 3.2. Impact of Renal Dysfunction on Overall and Cardiovascular Mortality

Clinical outcomes were available at 30 days for all patients. Long-term follow-up was available in 584 patients (86.1%) with a median follow-up of 659 (382–993) days. Compared with those with preserved eGFR, patients with RD had significantly higher 30-day mortality, one-year mortality, overall mortality, and all-cause mortality (Table 6, Figure 2A,B). The occurrence of AKI was higher in patients with RD although the difference among subgroups did not reach statistical significance.

### 3.3. Predictors of 30-Day Mortality

In the entire cohort, 30-day mortality was 2.8% (19 patients). Mortality rate was 1.8% in patients with preserved eGFR, 4.7% in patients with eGFR 45–60 mL/min/1.73 m^2^ and of 6.5% in patients with eGFR < 45 mL/min/1.73 m^2^ (*p* = 0.030). Thirty-day mortality was 1.8% in patients with sPESI of 0 and of 3.8% in patients with sPESI ≥ 1 (*p* = 0.118). By univariate Cox analysis, age, active malignancy, systolic blood pressure, renal dysfunction at diagnosis, BNP and CRP level were significant predictors of 30-day mortality. By multivariate analysis, renal dysfunction at diagnosis and elevated CRP levels were independent predictors of 30-day mortality (Table 3).

### 3.4. Predictors of Overall Mortality during Follow-Up

One hundred and seven deaths (107/584, 18.3%) were observed during a median follow-up of 659 (382–993) days. Mortality rate was of 14.4% in patients with preserved eGFR, 22.6% in patients with eGFR 45–60 mL/min/1.73 m^2^ and of 36.7% in patients with eGFR < 45 mL/min/1.73 m^2^ (*p* < 0.001). By univariate Cox analysis, age, sPESI, active malignancy, history of arterial hypertension, RD at diagnosis, AKI during hospital stay, BNP levels, PaO_2_, decrease in left ventricular ejection fraction (LVEF) elevated pulmonary artery pressure, were significant predictors of overall mortality. By multivariate analysis, sPESI, RD at diagnosis, AKI during hospital stay remained independent predictors of overall mortality (Table 4).

### 3.5. Predictors of Cardiovascular Mortality during Follow-Up

Cardiovascular death occurred in twenty-four patients during follow-up. Cardiovascular mortality rate was 2.3% in patients with preserved eGFR, 5.7% in patients with eGFR 45–60 mL/min/1.73 m^2^ and 15.9% in patients with eGFR <45 mL/min/1.73 m^2^ (*p* < 0.001). By univariate Cox analysis, age, sPESI, RD at diagnosis, BNP levels, hypoxemia, were significant predictors of cardiovascular mortality. By multivariate analysis, RD at diagnosis was the sole independent predictor of cardiovascular death (Table 5).

### 3.6. Renal Dysfunction and Simplified Pulmonary Embolism Severity Index (sPESI) for Prognostic Assessment of Acute Pulmonary Embolism (APE) Patients

Data was further analyzed to investigate the relationship between renal dysfunction and sPESI in APE patients. Group 1 comprised 271 patients with preserved eGFR at diagnosis and sPESI at 0, Group 2 comprised 285 patients with either eGFR <60 mL/min/1.73 m^2^ or sPESI ≥1, Group 3 comprised 121 patients with eGFR <60 mL/min/1.73 m^2^ and sPESI ≥1. A significant stepwise increase in AKI, 30-day mortality, one-year mortality, all-cause mortality and cardiovascular mortality was observed between these groups (Table 7, Figure 3A,B), with the highest mortality noted in the group with RD and elevated sPESI.

## 4. Discussion

The main findings of the present study are (i) the presence of RD in APE patients is associated with significantly increased 30-day mortality, all-cause mortality and cardiovascular mortality during follow-up, (ii) the addition of RD at diagnosis to the sPESI improves risk stratification and identifies a subset of patients with high mortality risk, and (iii) the study confirms the high prevalence of RD in patients with APE.

### 4.1. High Prevalence of Renal Dysfunction in APE Patients

Recent findings from a large nationwide dataset have underlined an increased incidence of APE with declining renal function. In 32,616,411 adult discharges from hospitals covered by the NIS (Nationwide Inpatient Sample) 2007 database in the Unites States [9], the annual frequency of PE was 527 per 100,000, 204 per 100,000, and 66 per 100,000 persons with end stage renal disease, CKD, and normal renal function, respectively. In accordance with this finding, other studies have reported a high proportion (up to 48%) of renal dysfunction as defined by eGFR <60 mL/min/1.73 m^2^ in patients with APE [7,8]. In comparison, the proportion of patients with eGFR <60 mL/min/1.73 m^2^ appears to be lower in the present study (26.9%). Additionally, the 30-day mortality rate observed in our cohort was lower than those reported in previous studies [7,14]. Possible explanations include the recruitment of a lower risk population in the present study.

### 4.2. Impact of Renal Dysfunction on Mortality in APE Patients

In our study, the increase in 30-day mortality, overall and cardiovascular mortality in patients with RD is in line with previous studies that have reported a negative association between previous RD or AKI and adverse outcomes [7,9,15,16,17,18]. In the International Cooperative Pulmonary Embolism Registry (ICOPER), elevated creatinine concentration (>177 mol/L) was found to be a predictor of mortality at 3 months [15]. Likewise, in the large NIS database [9], in-hospital mortality was doubled in CKD patients or patients with end stage renal failure compared to patients with normal renal function. Other studies have highlighted the accuracy of eGFR in predicting mortality when coupled together with troponin elevation, heart rate, and the presence of congestive heart failure [7].

In the setting of APE, RD can be due to previous underlying CKD and/or de novo impairment of renal function with APE associated hemodynamic insults. Another possible contributor of AKI in the setting of APE is contrast-induced nephropathy. Incidence of contrast-induced nephropathy in patients referred for CT pulmonary angiography for suspected APE was reported to range between 8.1% and 12% [19,20,21]. Independent predictors for contrast-induced nephropathy in this setting include age >75 years, diabetes mellitus, use of nonsteroidal anti-inflammatory drugs and multiple myeloma [19]. In our study, independent predictors of AKI were sPESI and interestingly, the use of ventilation/perfusion scans as the diagnostic method. This apparent paradox likely reflects our clinical practice, where CT pulmonary angiography was preferentially used in patients deemed less likely to develop CIN (younger age and preserved renal function).

Previous reports have emphasized that eGFR measured during the acute phase could be a stronger mortality predictor than history of previous kidney disease, which suggests that AKI may contribute per se to the pathogenesis of APE [7]. In our study, the proportion of AKI appears to be lower (18.8%) than previously described (30%) [8]. This difference could be attributed to the inclusion of lower-risk patients as evidenced by the lower 30-day mortality observed in our cohort (2.8% vs. 10%) [8]. In an attempt to decipher the respective contribution of AKI and previous CKD on adverse outcomes in APE patients, Kostrubiec and coworkers studied the prognostic impact of neutrophil gelatinase-associated lipocalin (N-GAL), a marker of tubular lesion and acute injury, cystacine C, an early marker of impaired glomerular filtration reflecting previous and acute renal dysfunction and eGFR. Whilst all of these markers were found to be predictive of 30-day mortality by univariate analysis, multivariate analysis revealed cystatin C as the strongest renal biomarker in the prediction of adverse outcomes [8]. In our study, RD at diagnosis was shown to be an independent predictor of 30-day mortality, overall mortality and cardiovascular mortality, whereas AKI was only shown to be an independent marker of overall mortality. Taken together, these data suggest that previous CKD and, perhaps to a lower extent, de novo AKI contribute independently to adverse outcomes in APE.

### 4.3. Assessment of Renal Function Improves sPESI Risk Stratification

Whereas the major strength of sPESI lies in excluding an adverse outcome in patients with sPESI of 0, the accuracy of sPESI ≥1 in risk assessment remains questionable. In the original description of the sPESI score, a 10-fold elevation of the mortality rate was demonstrated in high-risk patients (sPESI ≥1: 10.9%) compared to low-risk patients (sPESI = 0: 1%) [5]. In the RIETE validation cohort, 30-day mortality of low risk and high risk patients were 1.1% and 8.9% respectively [5]. The low 30-day mortality rate observed in the present study in patients with a sPESI score of 0 (1.8%) is consistent with the original description by Jimenez and in line with the first report using the PESI score (1.6%) [22]. In contrast, the impact of sPESI ≥ 1 in risk assessment appears less pronounced in the present cohort (30-day mortality rate of 3.8%). Our data demonstrates the limited value of sPESI in risk stratification of APE patients when renal dysfunction is not taken into account. In our study, the sPESI by itself did not appear to be an independent predictor of 30-day and cardiac mortality.

In an attempt to overcome the limited accuracy of current risk scores including the sPESI, attention was shifted to prognostic models combining clinical, imaging, and biochemical parameters. Some cohort studies have highlighted that biomarkers may confer prognostic value in addition to clinical parameters and echocardiography [1]. Compared to complex multimodal approaches for risk stratification, the assessment of renal dysfunction by eGFR offer several key advantages including wide availability, simplicity and reproducibility. As declining renal function tends to co-segregate with other classic cardiovascular risk factors, eGFR can be viewed as an integrated marker reflecting not only previous CKD damage but also comorbid conditions. In addition, changes in eGFR alteration could also indicate acute hemodynamic disturbances associated with APE. In regard to the sPESI, the additional value of RD in risk stratification in APE is illustrated in Figure 3A,B and Table 7. Acute kidney injury, 30-day mortality, one year mortality, overall mortality and cardiovascular mortality were the highest in patients with RD and sPESI ≥1. Although the present study was not designed to provide mechanistic insights explaining the adverse outcomes in patients with RD, several hypotheses could be raised. The mechanisms through which renal dysfunction affects the clinical outcome are likely to be multifactorial and include endothelial dysfunction, persistent micro-inflammation, an ongoing procoagulant state, increased bleeding risk or insufficient use of well-proven therapies.

### 4.4. Study Limitations

The present study was a single center retrospective study of consecutive patients with APE. Echocardiographic parameters to assess for right ventricular dysfunction were not systematically studied. Although the cardiovascular events were not adjudicated by an independent committee, this adjudication was performed by two physicians (M.G. and O.M.) who were blinded to the eGFR and sPESI values. Given the relatively low number of cardiac deaths recorded in this registry, multivariate analyses should be interpreted with caution and the findings viewed as exploratory and/or hypothesis generating. Moreover, our findings have not been replicated in an external validation cohort. As with similar evaluation of registry data, there are inherent limitations to this type of study, mainly related to known or unknown confounding factors. Finally, our study used the MDRD formula in order to estimate eGFR although the CKD-EPI formula showed its superiority in analyzing the prognosis of normotensive patients with APE in a previous study [23].

## 5. Conclusions

In patients’ APE, RD is associated with increased 30-day mortality, all-cause mortality and cardiovascular mortality. Implementation of renal dysfunction as assessed by eGFR <60 mL/min/1.73 m^2^ on admission to the sPESI score identifies a specific subset of patients at very high mortality risk.

## Figures and Tables

**Figure 1 jcm-08-00160-f001:**
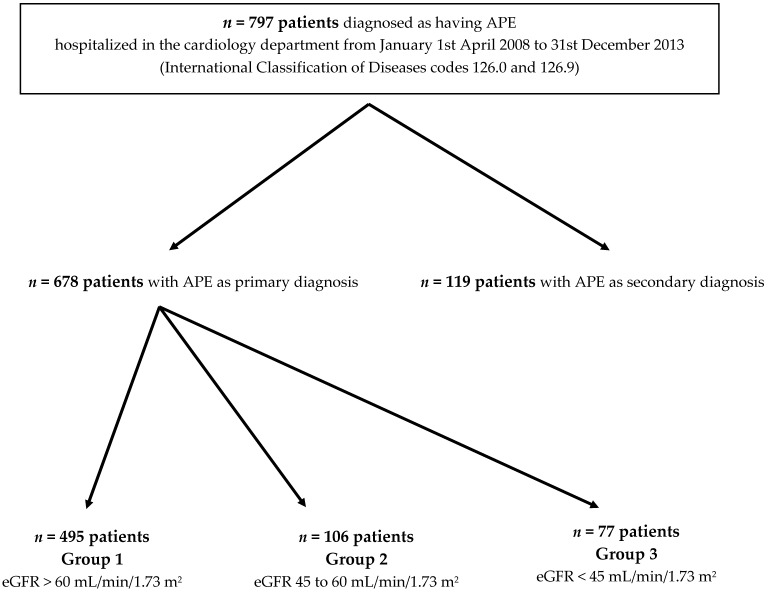
Flow chart. *n*: number.

**Figure 2 jcm-08-00160-f002:**
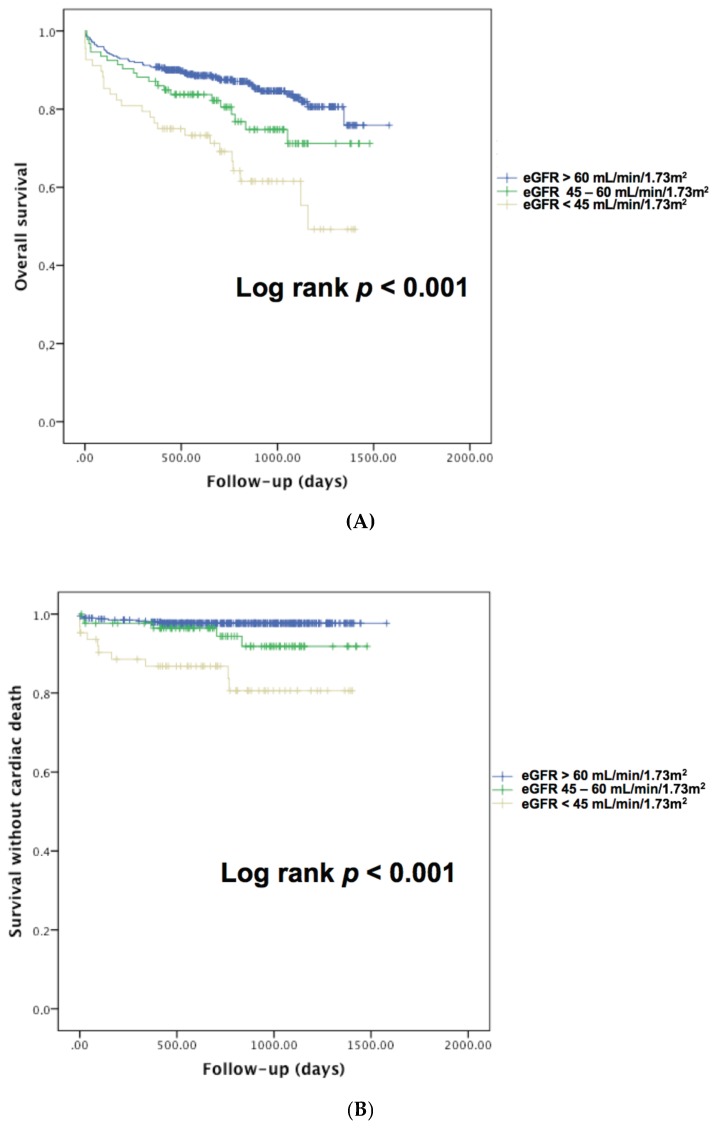
(**A**) Kaplan–Meier analysis for the probability of overall survival according to renal dysfunction in patients with acute pulmonary embolism. (**B**) Kaplan–Meier analysis for the probability of cardiac survival according to renal dysfunction in patients with acute pulmonary embolism.

**Figure 3 jcm-08-00160-f003:**
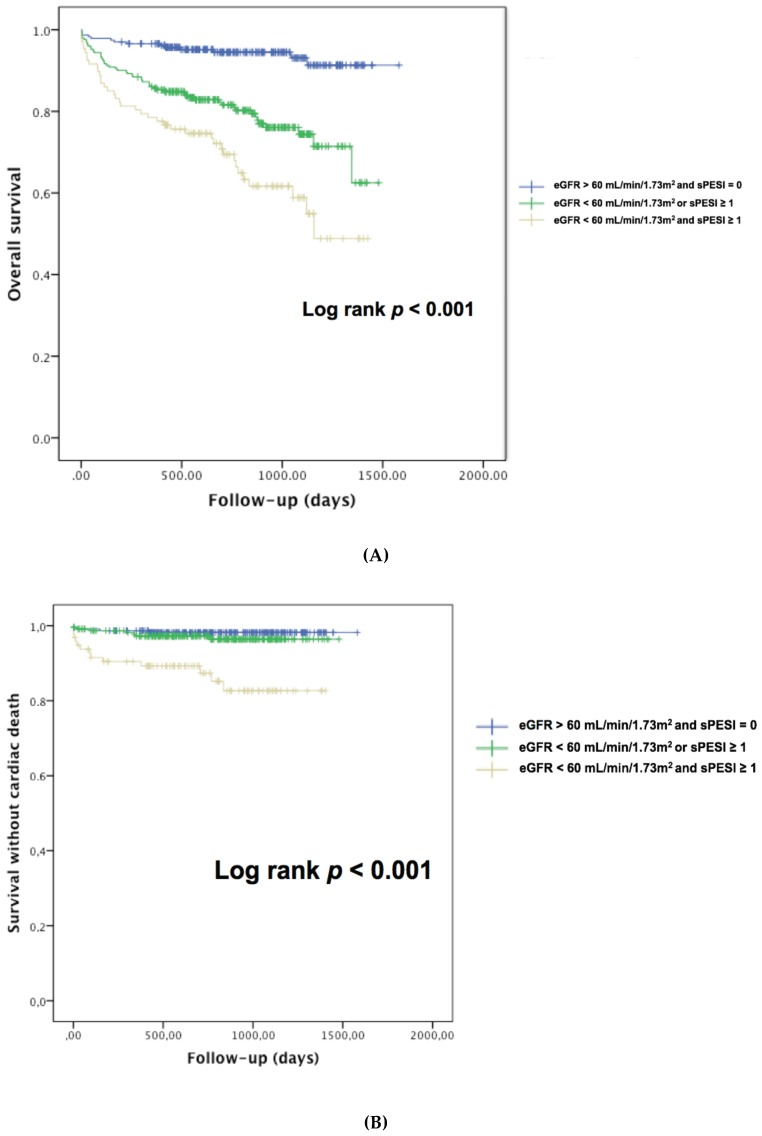
(**A**) Kaplan–Meier analysis for the probability of overall survival according to sPESI and renal dysfunction in patients with acute pulmonary embolism. (**B**) Kaplan–Meier analysis for the probability of cardiac survival according to sPESI and renal dysfunction in patients with acute pulmonary embolism.

**Table 1 jcm-08-00160-t001:** Baseline clinical and echocardiographic characteristics.

	Group 1 (gr1)	Group 2 (gr2)	Group 3 (gr3)	*p*
	eGFR > 60 mL/min/1.73 m^2^ (*n* = 495)	eGFR 45–60 mL/min/1.73 m^2^ (*n* = 106)	eGFR <45 mL/min/1.73 m^2^ (*n* = 77)	
Age (yr)	62 ± 18	78 ± 10	77 ± 11	<0.001 ^#,^ ^β^
Female Gender	274 (55.4)	62 (58.5)	48 (62.3)	0.473
Weight (kg)	82 ± 20	77±17	81±18	0.069
Height (cm)	169 ± 11	166 ± 8	165 ± 8	0.006 ^β^
Systolic Arterial pressure (mmHg)	132 ± 23	133 ± 24	131 ± 24	0.823
Diastolic arterial pressure (mmHg)	75 ± 13	74 ± 13	72 ± 12	0.249
Heart rate (bpm)	89 ± 25	90 ± 22	88 ± 22	0.882
Malignancy	62 (12.6)	17 (16)	6 (7.8)	0.252
Diabetes Mellitus	70 (14.2)	18 (17)	23 (30.3)	0.002
Dyslipidemia	160 (32.5)	44 (41.5)	37 (48.1)	0.011
Arterial hypertension	227 (46)	80 (75.5)	62 (80.5)	< 0.001
Smoker	132 (26.8)	27 (25.5)	23 (29.9)	0.795
sPESI	0.52 ± 0.65	0.81 ± 0.64	0.74 ± 0.67	<0.001
Creatine at diagnosis (μmol/L)	69 ± 16	102 ± 15	162 ± 65	<0.001 ^#,^ ^β^
eGFR at diagnosis (mL/min/1.73 m^2^)	97 ± 8	53 ± 8	33 ± 8	<0.001 ^#,^ ^β^
Troponine (μg/L)	0.85 ± 0.8	0.60 ± 1	0.87 ± 2	0.956
BNP (ng/L)	235 ± 377	314 ± 296	561 ± 736	<0.001 ^β^
CRP (mg/L)	50 ± 75	53 ± 56	46 ± 5	0.294
Haemoglobin (g/dL)	13.2 ± 1.8	13.0 ± 1.7	12.8 ± 1.7	0.175
PaO_2_ (mmHg)	77 ± 24	83 ± 40	79 ± 27	0.180
PaCO_2_ (mmHg)	33 ± 6	33 ± 5	33 ± 7	0.630
SaO_2_ (%)	94 ± 4	93 ± 4	93 ± 6	0.149
D-Dimers (μg/L)	6832 ± 6268	7865 ± 6878	8010 ± 6925	0.226
LVEF (%)	59 ± 9	58 ± 9	54 ± 13	0.001 ^β^
Systolic PAP	40 ± 15	44 ± 13	45 ± 16	0.018 ^β^

sPESI: simplified pulmonary embolism severity index; BNP: brain natriuretic peptide; eGFR: estimated glomerular filtration rate; CRP: C-reactive protein; LVEF: left ventricular ejection fraction; PAP: pulmonary artery pressure. # gr1 vs. gr2 < 0.05; β gr1 vs. gr3 < 0.05. The proportion of missing value was less than 10% except for BNP (18.2%); PaO_2_ (25.6%), PaCO_2_ (25.7%); D-dimers (20%); Systolic Pulmonary Arterial pressure: 34%.

**Table 2 jcm-08-00160-t002:** Univariate and multivariate analyses for the prediction of acute kidney injury during hospital stay.

	Univariate Analysis			Multivariate Analysis		
	Hazard Ratio (HR)	95% CI	*p*	Hazard Ratio (HR)	95% CI	*p*
Age (years)	1.028	1.015–1.041	<0.001			
**sPESI**	**1.401**	**1.063–1.846**	**0.017**	**1.359**	**1.019–1.814**	**0.037**
Overweight	0.919	0.590–1.431	0.708			
SBP (mmHg)	1.003	0.994–1.011	0.517			
DBP (mmHg)	1.003	0.988–1.017	0.723			
Heart rate (bpm)	1.008	0.999–1.017	0.082			
Acitve malignancy	0.906	0.500–1.642	0.746			
Diabetes mellitus	1.682	1.045–2.707	0.032	1.314	0.796–2.171	0.286
Dyslipidemia	1.104	0.741–1.644	0.628			
Arterial Hypertension	1.621	1.089–2.414	0.017	1.325	0.867–2.025	0.194
Smoker	1.027	0.666–1.582	0.905			
RD at diagnosis (eGFR < 60 mL/min/1.73 m^2^)	1.417	0.935–2.147	0.101			
**CT scan**	**0.365**	**0.242–0.551**	**<0.001**	**0.394**	**0.258–0.600**	**<0.001**
Troponine (μg/L)	0.998	0.969–1.027	0.870			
BNP > 400 ng/L	1.779	1.113–2.854	0.013			
CRP (mg/L)	1.000	0.997–1.003	0.853			
PaO2 (mmHg)	1.003	0.996–1.010	0.421			
LVEF < 50%	2.108	1.290–3.446	0.003			
ACE + sartans	1.373	0.921–2.047	0.119			

SBP: systolic blood pressure: DBP: diastolic blood pressure; RD: renal dysfunction; CI: confidence interval. Variables that were taken into account in the sPESI (age, cancer, chronic heart failure or chronic pulmonary disease, systolic blood pressure < 100 mmHg, arterial oxyhemoglobinsaturation < 90%) or clearly related to the sPESI score (BNP, LVEF < 50%, PaO_2_) were not entered into the multivariate analysis model.

**Table 3 jcm-08-00160-t003:** Univariate and multivariate analyses for the prediction of 30-day mortality.

	Univariate Analysis			Multivariate Analysis		
	Hazard Ratio (HR)	95% CI	*p*	Hazard Ratio (HR)	95% CI	*p*
Age (years)	1.040	1.002–1.079	0.036			
sPESI	1.545	0.824–2.896	0.175	1.453	0.735–2.871	0.282
SBP (mmHg)	0.973	0.948–0.997	0.034			
DBP (mmHg)	0.970	0.928–1.014	0.174			
Heart rate (bpm)	1.015	0.992–1.038	0.208			
Acitve malignancy	3.149	1.094–9.064	0.033			
Diabetes mellitus	0.716	0.163–3.151	0.659			
Dyslipidemia	0.600	0.194–1.861	0.377			
Arterial Hypertension	1.410	0.512–3.879	0.506			
Smoker	0.901	0.291–2.795	0.857			
**RD at diagnosis** **(eGFR < 60 mL/min/1.73 m^2^)**	**2.742**	**1.029–7.305**	**0.044**	**2.771**	**1.007–7.625**	**0.048**
AKI	1.948	0.677–5.606	0.216			
CT Scan	0.792	0.261–2.407	0.681			
Troponine (μg/L)	0.998	0.920–1.081	0.952			
BNP > 400ng/L	2.894	1.004–8.342	0.049			
**CRP (mg/L)**	**1.003**	**1.001–1.005**	**0.011**	**1.004**	**1.001–1.006**	**0.004**
PaO2 (mmHg)	1.008	0.995–1.022	0.204			
LVEF <50%	1.337	0.284–6.297	0.713			

AKI: acute kidney injury. Variables that were taken into account in the sPESI (age, cancer, chronic heart failure or chronic pulmonary disease, systolic blood pressure < 100 mmHg, arterial oxyhemoglobinsaturation < 90%) or clearly related to the sPESI score (BNP, LVEF < 50%, PaO_2_) were not entered into the multivariate analysis model.

**Table 4 jcm-08-00160-t004:** Univariate and multivariate analyses for the prediction of overall mortality.

	Univariate Analysis			Multivariate Analysis		
	Hazard Ratio (HR)	95% CI	*p*	Hazard Ratio (HR)	95% CI	*p*
Age (years)	1.040	1.029–1.059	<0.001			
**sPESI**	**2.057**	**1.672–2.531**	**<0.001**	**1.895**	**1.520–2.362**	**0.001**
SBP (mmHg)	0.991	0.982–1.000	0.039			
DBP (mmHg)	0.983	0.967–0.998	0.030			
Heart rate (bpm)	1.005	0.996–1.013	0.308			
Acitve malignancy	4.402	2.945–6.581	<0.001			
Diabetes mellitus	1.042	0.628–1.729	0.874			
Dyslipidemia	1.142	0.774–1.683	0.504			
Arterial Hypertension	1.936	1.279–2.929	0.002	1.284	0.823–2.003	0.270
Smoker	0.967	0.628–1.488	0.878			
**RD at diagnosis (eGFR < 60 mL/min/1.73 m^2^)**	**2.151**	**1.467–3.155**	**<0.001**	**1.772**	**1.188–2.645**	**0.005**
**AKI**	**2.101**	**1.398–3.157**	**<0.001**	**1.655**	**1.091–2.510**	**0.018**
Troponine (μg/L)	1.001	0.977–1.026	0.916			
BNP > 400 ng/L	2.235	1.471–2.235	<0.001			
CRP (mg/L)	1.002	1.000–1.003	0.073			
PaO2 (mmHg)	1.010	1.005–1.016	<0.001			
LVEF < 50%	2.603	1.679–4.035	<0.001			

Variables that are taken into account in the sPESI (age, cancer, chronic heart failure or chronic pulmonary disease, systolic blood pressure < 100 mmHg, arterial oxyhemoglobinsaturation < 90%) or clearly related to the sPESI score (BNP, LVEF < 50%, PaO_2_) were not enter into the multivariate analysis model.

**Table 5 jcm-08-00160-t005:** Univariate and multivariate analyses for the prediction of cardiovascular mortality.

	Univariate Analysis			Multivariate Analysis		
	Hazard Ratio (HR)	95% CI	*p*	Hazard Ratio (HR)	95% CI	*p*
Age (years)	1.067	1.028–1.106	<0.001			
sPESI	1.760	1.093–2.833	0.020	1.607	0.946–2.733	0.080
SBP (mmHg)	0.994	0.975–1.014	0.563			
DBP (mmHg)	0.990	0.956–1.025	0.563			
Heart rate (bpm)	1.009	0.991–1.028	0.338			
Acitve malignancy	0.799	0.188–3.401	0.762			
Diabetes mellitus	1.664	0.660–4.191	0.280			
Dyslipidemia	0.933	0.453–2.367	0.933			
Arterial Hypertension	1.427	0.625–3.262	0.339			
Smoker	1.351	0.578–3.158	0.487			
**RD at diagnosis** **(eGFR < 60 mL/min/1.73 m^2^)**	**4.718**	**2.064–10.782**	**<0.001**	**4.246**	**1.856–9.710**	**0.001**
AKI	2.283	0.977–5.336	0.057	1.953	0.835–4.569	0.123
Troponine (μg/L)	1.002	0.959–1.047	0.926			
BNP > 400ng/L	4.791	2.033–11.292	<0.001			
CRP (mg/L)	1.002	0.999–1.005	0.123			
PaO2 (mmHg)	1.015	1.007–1.024	0.001			
LVEF <50%	1.483	0.495–4.440	0.481			

Variables that were taken into account in the sPESI (age, cancer, chronic heart failure or chronic pulmonary disease, systolic blood pressure < 100 mmHg, arterial oxyhemoglobinsaturation < 90%) or clearly related to the sPESI score (BNP, PaO_2_) were not entered into the multivariate analysis model.

**Table 6 jcm-08-00160-t006:** Events according to renal dysfunction. 30-day mortality was based on the whole study population (group 1 *n* = 495, group 2 *n* = 106, group 3 *n* = 77).

	Group 1 (gr1)	Group 2 (gr2)	Group 3 (gr3)	*p*
	eGFR >60 mL/min/1.73 m^2^ (*n* = 423)	eGFR 45–60 mL/min/1.73 m^2^ (*n* = 93)	eGFR <45 mL/min/1.73 m^2^ (*n* = 68)	
Acute kidney injury, *n* (%)	86 (17.4)	20 (18.9)	22 (28.6)	0.065
30 days mortality, *n* (%)	9 (1.8)	5 (4.7)	5 (6.5)	0.030
One year mortality, *n* (%)	42 (9.9)	14 (15.1)	17 (25)	0.002
All cause mortality, *n* (%)	61 (14.4)	21 (22.6)	25 (36.8)	<0.001
Cardiovascular mortality, *n* (%)	9 (2.3)	5 (5.7)	10 (15.9)	<0.001
Follow-up (days)	670 ± 405	652 ± 397	600 ± 417	0.372

**Table 7 jcm-08-00160-t007:** Events according to renal dysfunction on admission and sPESI score.

	eGFR > 60 and sPESI = 0 (*n* = 271)	eGFR < 60 or sPESI ≥ 1 (*n* = 285)	eGFR < 60 and sPESI ≥ 1 (*n* = 121)	*p*
Acute kidney injury, *n* (%)	38 (14)	61 (21.4)	29 (24)	0.025
30 day mortality, *n* (%)	4 (1.5)	7 (2.5)	8 (6.6)	0.016
One year mortality, *n* (%)	10 (4.4)	38 (15.3)	25 (23.4)	<0.001
All cause mortality, *n* (%)	14 (6.1)	54 (21.8)	39 (36.4)	<0.001
Cardiovascular mortality, *n* (%)	4 (1.8)	7 (3.1)	13 (13.3)	< 0.001

eGFR is expressed in mL/min/1.73 m^2^.

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
