# Peer review of "Assessment of Renal Dysfunction Improves the Simplified Pulmonary Embolism Severity Index (sPESI) for Risk Stratification in Patients with Acute Pulmonary Embolism"

_jcm, 2019, doi:10.3390/jcm8020160_

Round 1
Reviewer 1 Report
Trimaille et al., performed a prospective study in 678 consecutive patients with acute pulmonary embolism (APE) to determine whether renal dysfunction assessment by eGFR improves the simplified pulmonary embolism severity index (sPESI) score risk stratification in patients with APE. The authors found that in patients with APE, the addition of RD to the sPESI score identifies a specific subset of patients at very high mortality.
Overall the study question is appropriate, and methods are sound. Authors have a done a great job in preparing the manuscript, describing the results and presenting the discussion.
Following are my comments and suggestions to the authors,
1.Introduction:
-Page 2 of 15. Line 612. Authors wrote ‘In addition, the impact of acute kidney injury defined as a > 25% increase in creatinine levels during hospital stay on adverse outcome was investigated.
Why was this definition of Acute kidney injury (AKI) was chosen when there’s a standard KDIGO- AKI definition, (1)?
-Page 3 of 15. Line 87, Authors wrote ‘For each patient, the type of renal failure (acute vs. chronic) was established by a careful reviewing of medical and biological records.
Need clarification. How were the patients diagnosed as having Chronic Kidney disease? What are all the criteria looked into?
2.Discussion:
4.4 Study limitations: Can authors comment on the use of MDRD for eGFR when compared to eGFR estimation with CKD-EPI equation would have made any difference to the study findings? (2)
References:
1. KDIGO Clinical Practice Guideline for Acute Kidney Injury, Kidney Int Suppl. 2012;2(Suppl 1):8.
2. Altınsoy, Bülent, İbrahim İlker, Tacettin Örnek, et al., “Prognostic Value of Renal Dysfunction Indicators in Normotensive Patients With Acute Pulmonary Embolism.” Clinical and Applied Thrombosis/Hemostasis, (September 2017), 554–61. doi:10.1177/1076029616637440.
Author Response
Point 1: Page 2 of 15. Line 612. Authors wrote ‘In addition, the impact of acute kidney injury defined as a > 25% increase in creatinine levels during hospital stay on adverse outcome was investigated. Why was this definition of Acute kidney injury (AKI) was chosen when there’s a standard KDIGO- AKI definition (ref 1)?
As pointed out by the Reviewer, the definition of AKI may appear arbitrary and simple at first glance.
The International Society of Nephrology/KDIGO has defined AKI as any of the following: (i) Increase in serum creatinine level (SCr) by ≥0.3 mg/dl (≥26.5 umol/l) within 48 hours; (ii) or increase in SCr to ≥1.5 times baseline, which is known or presumed to have occurred within 7 days, (iii) or Urine volume < 0.5 ml/kg/h for 6 hours.
As mentioned by the Reviewer, the KDIGO- AKI definition should remain the standard when assessing AKI and enables reproducibility, uniformity and accuracy amongst AKI related literature. Nevertheless, we thought that this definition might be too strict and suffer from a lack of sensitivity when initially conducting the design of our study. Moreover, the visual and easy to calculate “cut off value of >25%” was first established to potentially develop an extended scoring system and further implement validated algorithm aimed to asses prognosis amongst PE patients. This threshold is also commonly used in other clinical settings to study the occurrence of contrast-induced nephropathy following percutaneous coronary intervention (Mehran R, Aymong ED, Nikolsky E et al. A simple risk score for prediction of contrast-induced nephropathy after percutaneous coronary intervention: development and initial validation. J Am Coll Cardiol 2004; 44: 1393-9 or Ivanes F, Isorni MA, Halimi JM et al. Predictive factors of contrast-induced nephropathy in patients undergoing primary coronary angioplasty. Arch Cardiovasc Dis 2014; 107: 424-32).
Point 2 : Page 3 of 15. Line 87, Authors wrote ‘For each patient, the type of renal failure (acute vs. chronic) was established by a careful reviewing of medical and biological records. Need clarification. How were the patients diagnosed as having Chronic Kidney disease? What are all the criteria looked into?
As pointed out by the Reviewer, the type of renal failure (acute vs chronic) faced important limitations due to the heterogeneity of our PE cohort (ii) incomplete medical history, (ii) lack or no prior documentation of kidney disease, (iii) acute renal failure complicating chronic kidney disease etc.
In order to reduce inherent bias due to the retrospective and observational design of our study, a careful reviewing of medical and biological records was performed by the authors. Pre existing chronic kidney disease (CKD) was determined at the time of enrolment for each patient: (i) according to current KDIGO definitions: kidney damage or glomerular filtration rate (GFR)<60 mL/min/1.73/m2 for 3 months or more, irrespective of cause (ii) by evaluating available clinical and laboratory data from the patient or surrogate (including general practitioner) (iii) using serum levels of calcium, phosphate, PTH, alkaline phosphatase activity and albuminuria when available to better discriminate between acute and chronic renal failure and (iv) eventually clinical judgment was required for some cases in order to determine if patients were likely to have AKI or CKD if incomplete clinical data were available to apply the diagnostic criteria.
Changes: To take into account the Reviewer’s comment, the sentence concerning the definition of chronic kidney disease has been changed:
” For each patient, the type of renal failure (acute vs chronic) was established by a careful reviewing of medical and biological records. Pre existing chronic kidney disease (CKD) was determined at the time of enrolment for each patient: (i) according to current KDIGO definitions: kidney damage or glomerular filtration rate (GFR)<60 mL/min/1.73/m2 for 3 months or more (ii) by evaluating available clinical and laboratory data from the patient or surrogate (iii) using serum levels of calcium, phosphate, parathormone, alkaline phosphatase activity and albuminuria when available”.
Point 3 : Study limitations: Can authors comment on the use of MDRD for eGFR when compared to eGFR estimation with CKD-EPI equation would have made any difference to the study findings? (ref 2)
We thank the Reviewer for his comment. An additional and extended work based on the current and proposed manuscript would be to (i) assess the optimal threshold of SCr increase associated with dismal prognosis amongst PE patients (ii) compare Cockcroft-Gault, Modification of Diet in Renal Disease (MDRD) and Chronic Kidney Disease Epidemiology Collaboration (CKD-EPI) baseline eGFR estimation to assess incidence, predictors and impact of baseline renal function amongst PE patients and (iii) finally assess the impact of AKI occurrence (early vs mid or late AKI).
To accommodate the Reviewer comment, we have added a statement in the limitations part regarding the equation used to estimate the glomerular filtration rate :
“Finally, our study used the MDRD formula in order to estimate eGFR although CKD-EPI formula previously showed a superiority in analysing prognosis of normotensive APE patients [23].”
Reviewer 2 Report
The manuscript by Trimaille A. et al. entitled “Assessment of renal dysfunction improves the simplified pulmonary embolism severity index (sPESI) for risk stratification in patients with acute pulmonary embolism” is a well-done study focused on assessing the consequences of addition of renal dysfunction to the sPESI score; the study hypothesized that this would help to identify the group of patients with acute pulmonary embolism (APE) at very high mortality. Despite of the some limitations, which have been taken into consideration by authors, the study provides strong evidence that in acute pulmonary embolism, glomerular filtration rate (GFR) can be viewed as an integral marker reflecting kidney damage and comorbid conditions and alteration in hemodynamics associated with APE.
The authors calculated the estimated GFR (eGFR) using the MDRD formula, which is not considering the body weight, which can directly affect GFR. This could provide more accurate assessment of the kidney damage. It would be good if the authors provide these data.
Did the authors estimate creatinine clearance, which takes into consideration the body weight? Please provide these data, if possible.
The figures are very hard to read. It would be good to rescale the y axis to expand the data range and consider changing the color scheme.
Author Response
Point 1 : Did the authors estimate creatinine clearance, which takes into consideration the body weight? Please provide these data, if possible.
We thank the Reviewer for his comment. Unfortunately, we did not estimate creatinine clearance in the proposed file. As suggested by the referee, for subgroups of patients who are old, underweight or overweight, no clear-cut advice exists regarding which formula (the Cockcroft-Gault, MDRD and CKD-EPI) is best to use for optimal estimation of kidney function. Its generally admitted that CKD-EPI gives the best estimation of GFR, although its accuracy is close to that of the MDRD.
As shown in the manuscript :
(i) Age differences exist (table 1: group 1 62 ± 18 vs group 2 78 ± 10 vs group 77 ± 11, p=0,001)
(ii) Weight showed no significant difference amongst the 3 groups (table 1: group 1 82 ± 20 vs Group 2 77±17, Group 3 81±180 ; p=069)
(iii) and overweight was not associated in univariate analysis with a increased risk of AKI (table 2, p =0.708).
It would have been of interest to record body mass index and dismiss any doubts about the homogeneity of our 3 compared groups.
As related previously (reviewer 1); an additional and extended work based on the current and proposed manuscript would be to compare Cockcroft-Gault, Modification of Diet in Renal Disease (MDRD) and Chronic Kidney Disease Epidemiology Collaboration (CKD-EPI) baseline eGFR estimation to assess incidence, predictors and impact of baseline renal function amongst PE patients.
Point 2 : The figures are very hard to read. It would be good to rescale the y axis to expand the data range and consider changing the color scheme.
As requested by the editorial office, all figures and tables have been inserted into the main text and may have suffered from altered resolution. To accommodate the Reviewer comment, TIFF files can be transferred separately to afford modifications by the publication coordinator in accordance with the journal requirements.
Reviewer 3 Report
Trimaille A et.al., were evaluated the role of Renal disfunction (RD) assessed by estimated glomerular filtration rate (eGFR) in In Acute pulmonary embolism (APE). Authors analyzed the 678 pulmonary embolism (PE) patient’s data using hospital database and concluded that RD is associated with increased 30-day mortality and cardiovascular mortality In APE patients. Data is analyzed thoroughly and the design of the study (Ex: 119 secondary diagnosis patients out of 797 APE were excluded from this study and the authors clearly mentioned the limitations of the study) is appropriate enough to achieve the aim mentioned in manuscript. The manuscript is well-presented and has interesting observation which is beneficial to APE patients, clinicians and researchers in kidney, cardiovascular and pulmonary fields.
I had no specific suggestions for the authors with exception of the following:
The results presented in tables were well organized based on the purpose of the study but, authors should take great care in presenting the values in a clear manner. Ex:
· In page 9, line 202: PaO2, is typed as Pa0 (zero)2
· line 215: group 1 comprised 272 patients. But it is mentioned in in table 7 group1 n= 271. Also please check the sub groups n number mentioned in table 6 and in the text.
· Line 217: sub group sPESI ≥ whereas in table it is mentioned as >1
· Authors are advised to increase the series font size in figure2 and 3
· In table 7: if possible adjust the % and < symbols in the same row next to the text
Author Response
Point 1 : In page 9, line 202: PaO2, is typed as Pa0 (zero)2
Change has been made according to the Reviewer’s request : PaO2.
Point 2 : Line 215: group 1 comprised 272 patients. But it is mentioned in in table 7 group1 n= 271.
We are sorry for this typo occurring during the preparation of the manuscript. Change has been made in table 7 with 272 patients.
Point 3 : Also please check the sub groups n number mentioned in table 6 and in the text.
Table 6 highlights the events according to renal dysfunction. 30-day mortality analysis was based on the whole study population (group 1 n = 495, group 2 n = 106, group 3 n = 77) with a total of 678 patients. While clinical outcomes were available at 30 days for all patients (678patients), long-term follow-up was only available in 584 patients with is defined in table 6 as Group 1 n= 423; group 2= 93 and Group 3 n=68 (total of 584 patients).
Point 4 : Line 217: sub group sPESI ≥ whereas in table it is mentioned as >1
Changes have been made according to the Reviewer’s request with uniform values both in the text and table.
Point 5 : Authors are advised to increase the series font size in figure 2 and 3.
As requested by the editorial office, all figures and tables have been inserted into the main text and may have suffered from altered resolution. To accommodate the Reviewer comment, TIFF files can be transferred separately to afford modifications by the publication coordinator in accordance with the journal requirements.
Point 6 : In table 7: if possible adjust the % and < symbols in the same row next to the text.
Changes have been made according to the Reviewer’s request with uniform values both in the text and table.